# Shared structure facilitates working memory of multiple sequences

**Qiaoli Huang**[1,2,3,4]*, **Huan Luo**[1,2,3]*

[1]School of Psychological and Cognitive Sciences, Peking University, Beijing, China; [2]PKU-IDG/McGovern Institute for Brain Research, Peking University, Beijing, China; [3]Beijing Key Laboratory of Behavior and Mental Health, Peking University, Beijing, China; [4]Max Planck Institute for Human Cognitive and Brain Sciences, Leipzig, Germany

**Abstract** Daily experiences often involve the processing of multiple sequences, yet storing them challenges the limited capacity of working memory (WM). To achieve efficient memory storage, relational structures shared by sequences would be leveraged to reorganize and compress information. Here, participants memorized a sequence of items with different colors and spatial locations and later reproduced the full color and location sequences one after another. Crucially, we manipulated the consistency between location and color sequence trajectories. First, sequences with consistent trajectories demonstrate improved memory performance and a trajectory correlation between reproduced color and location sequences. Second, sequences with consistent trajectories show neural reactivation of common trajectories, and display spontaneous replay of color sequences when recalling locations. Finally, neural reactivation correlates with WM behavior. Our findings suggest that a shared common structure is leveraged for the storage of multiple sequences through compressed encoding and neural replay, together facilitating efficient information organization in WM.

*For correspondence:
qiaolihuang0818@gmail.com (QH);
huan.luo@pku.edu.cn (HL)

## eLife assessment

This **valuable** study uses a novel experimental design to elegantly demonstrate how we exploit stimulus structure to overcome working memory capacity limits. The presented behavioural and neural evidence are **solid** and in line with the proposed information compression mechanism. This study will be of interest to cognitive neuroscientists studying structure learning and memory.

## Introduction

A well-known feature of working memory (WM) is its limited capacity (*Baddeley, 2000*; *Cowan, 2001*), which constrains the amount of information that can be temporarily retained for future behavior. Meanwhile, in daily experiences, memorized items do not exist independently but are always part of a common framework or share the same structure, which could be leveraged to compress information and overcome the WM capacity challenge (*Brady et al., 2011*). For example, while shopping at a supermarket, numerous items could be grouped into a few categories, such as drinks, vegetables, fruits, and meats, to facilitate memory. Other types of abstract associations such as relational regularities and structure schema could also mediate WM organization (*Al Roumi et al., 2021*; *Gathercole and Baddeley, 2014*; *Mathy and Feldman, 2012*). Computational modeling also suggests that higher-order structures (e.g. summaries and relative relations) would reduce memory uncertainty by constraining individual-item representations (*Brady and Tenenbaum, 2013*; *Ding et al., 2017*).

**eLife digest** When we memorize a grocery list before heading into the store, we make use of our working memory. This type of neural process allows us to temporarily store the knowledge needed for a task, yet its capacity is limited. Having to recall more than one type of information at the same time, in particular, can quickly create challenges. Exactly how the brain maximizes the use of this limited working memory space remains unclear.

One possible strategy would be to take advantage of the patterns or connections that exist between seemingly unrelated pieces of information – for example, by remembering to buy apples, oranges and bananas under one broader 'fruit' category. To explore if this may be the case, Qiaoli Huang and Huan Luo designed a memory task in which two types of information were either connected through an underlying pattern (aligned trajectory condition) or completely independent (misaligned trajectory condition). Participants watched three colored dots appearing on screen one after the other, in such a way that they seemed to 'travel' around an imaginary circle. The volunteers were then asked to recall, in order, the location and color of each dot. Performance increased when color and location information were structured in the same way – that is, when both emerged from the three dots traveling around a circle or a color wheel with the same trajectory.

Recording the brain activity of the participants 'live' as they performed the task indicates that, in the aligned trajectory condition, the brain 'compresses' both types of information and extracts their common structure. Even when participants were asked to recall only the location of the dots, their brain also spontaneously replayed the related color information. Taken together, these findings provide new insights into how working memory aids in multitasking, a crucial aspect of our daily lives, and lay the groundwork for further exploration of this capability.

Cognitive maps, as one type of spatial schema (*Farzanfar et al., 2023*; *Gilboa and Marlatte, 2017*), provide a general structure framework for organizing information in different tasks and across various domains (*Tolman, 1948*; *Whittington et al., 2020*). They were first identified as representations of physical maps during navigation, but recently have been shown to also support other higher-level processes, such as conceptual knowledge, reasoning, planning, and decision-making (*Behrens et al., 2018*; *Bellmund et al., 2018*; *O'keefe and Nadel, 1978*). Two major neural signatures of cognitive maps, grid-like code (*Constantinescu et al., 2016*; *Doeller et al., 2010*; *Hafting et al., 2005*; *Park et al., 2021*), and neural replay in the hippocampal-entorhinal system (*Foster and Wilson, 2006*; *Liu et al., 2019*; *Liu et al., 2021b*; *Schuck and Niv, 2019*; *Skaggs and McNaughton, 1996*), are identified in both spatial and non-spatial tasks. Neural replay, the rapid item-by-item reactivation in a forward or backward direction, is posited to not only repeat past experiences, but also reflect an internal model of the world (*Kurth-Nelson et al., 2023*; *Ólafsdóttir et al., 2018*).

Accordingly, many higher-level processes could be described as mental explorations of a sequence of states within an abstract map, similar to tracing a route on a physical map. As a result, a cognitive map can serve as a common reference for aligning different features or domains. In line with the view, conjoined cognitive maps have been recently revealed in the rodent hippocampus (i.e. physical space and abstract task variables) (*Nieh et al., 2021*), and alignment of different feature maps speeds learning performance (*Aho et al., 2022*). Moreover, both spatial and conceptual distances are relied on to generalize when searching for correlated rewards in value-guided learning, supporting a cognitive-map-dependent computational mechanism (*Wu et al., 2020*). Based on these findings, we hypothesize that cognitive maps might be employed to reorganize memory information across domains to overcome capacity bottlenecks in WM.

Here, we examined whether cognitive maps shared by multiple feature domains would be naturally leveraged to make efficient use of limited WM capacity. To address this question, participants were asked to memorize a color sequence presented at a list of spatial locations and later reproduce both the color and location sequences on two rings, one after another. In other words, subjects need to retain in WM two sequences of features, i.e., color and location, both of which could be characterized as sequence trajectories along their respective rings on a spatial map. Crucially, we manipulate the consistency between color and location sequence trajectories on the rings. Specifically, for the aligned condition, the color and location sequence share a common spatial trajectory, i.e., separated by the

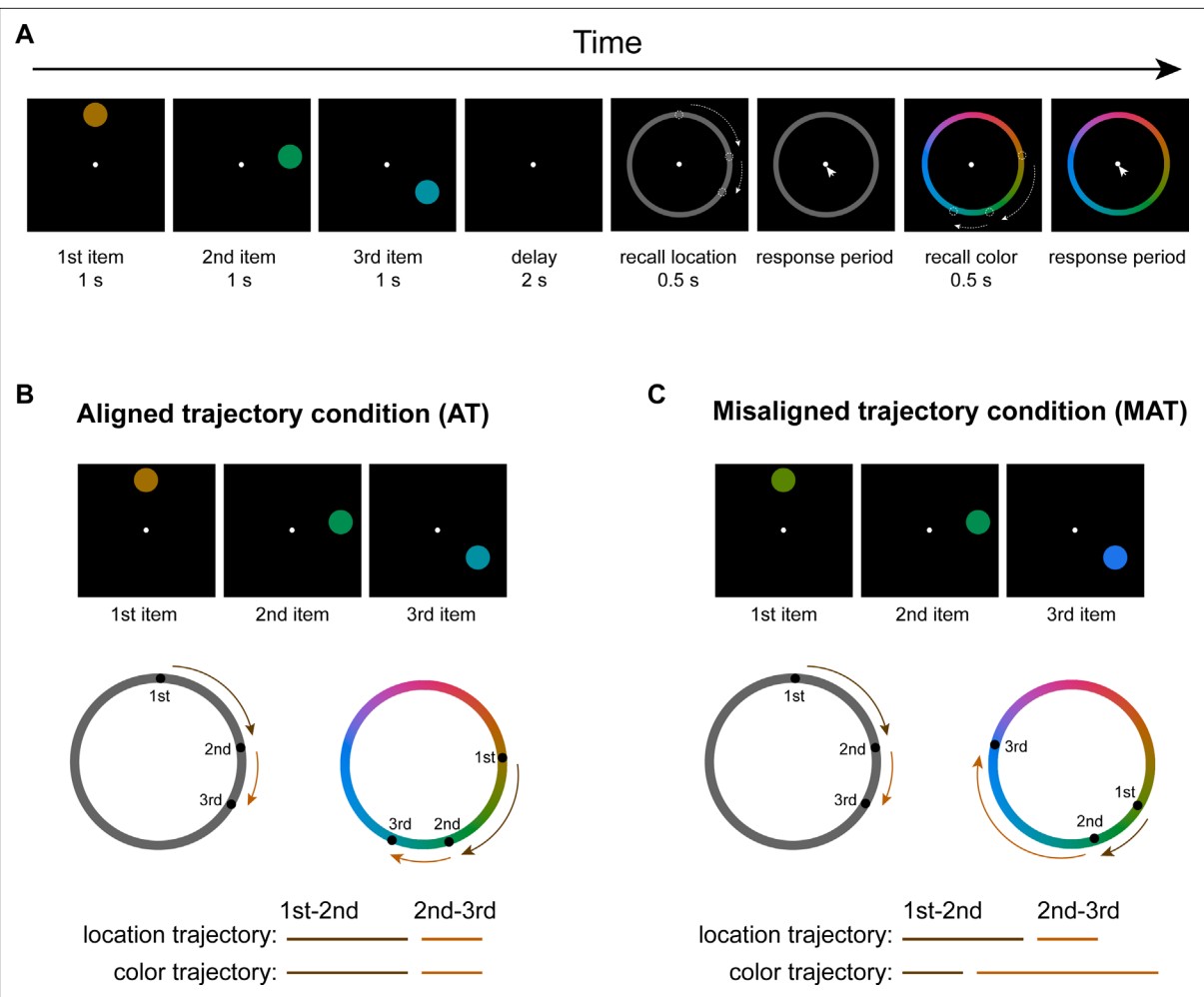

**Figure 1.** Experimental paradigm. (**A**) Participants were presented with a sequence of disks of different colors and at different locations. They were asked to memorize both the location and color of the sequence and later reproduce the full location and color sequences one after another by clicking the corresponding positions on the respective report rings. During the 'recall location' phase, a gray ring appeared and participants prepared for subsequent location recall without motor movement, to ensure memory decoding without motor interventions. During the following 'response period,' subjects serially selected memorized spatial locations on a 'location ring.' Next, a color ring appeared ('recall color') for subjects to be ready for subsequent color recall. They then clicked the remembered colors on a 'color ring' ('response period'). (**B**) Aligned trajectory condition (AT) wherein the trajectory distances between consecutive items (i.e. first to second, second to third) of location and color sequences are identical, although the two sequences occupy different locations within their respective rings. (**C**) Misaligned condition (MAT), wherein the trajectory distances between consecutive items are different for location and color sequences.

same distance between successive items between maps (*Figure 1B*), whereas, for the misaligned condition, they have distinct relative trajectories (*Figure 1C*). We hypothesize that humans would spontaneously combine the structure shared by the two sequences to facilitate memory formation, even though it is unsupervised and non-mandatory.

To preview the results, we provide converging behavioral and neural evidence for spontaneously leveraging common structures to facilitate WM. Behaviorally, sequences with consistent color-location trajectories (aligned condition) show enhanced memory precision and a significant correlation between reproduced color and location sequence trajectories. Neurally, aligned location-color sequences demonstrate reactivation of shared trajectory during both encoding and retention periods and interestingly, spontaneous replay of color sequences when recalling location sequences. Together, shared common structures enable the storage of multiple sequences in WM through compressed encoding and neural replay.

# Results

## Experimental procedure and behavior performance

Thirty-three human participants performed a visual sequence WM task while their brain activities were recorded using EEG. As shown in *Figure 1A*, at the beginning of each trial, three disks with different spatial locations and colors were sequentially presented. Participants were required to concurrently remember their locations and colors as well as their orders, i.e., one location sequence and one color sequence. After a 2 s memory delay, a gray ring (location ring) was presented to instruct participants to prepare for subsequent location sequence recall without making motor responses (*Figure 1A*, 'recall location' period). This is to ensure memory signals are decoded without explicit motor interventions. Next, a cursor appeared at the center of the screen ('response period'), and participants clicked the three spatial locations on a 'location ring' in their correct order. Upon completion of location recall, participants were instructed to prepare for color sequence recall ('recall color'), and they clicked three locations on the color ring based on the color sequence ('response period'). One key aspect was manipulating the consistency between location and color trajectories so that the two sequences share or do not share a common structure in a cognitive map. Specifically, in the aligned trajectory condition (*AT* condition), despite the location and color sequences occupying different positions within their respective rings, their trajectory distances (between first and second items and between second and third) were the same (*Figure 1B*). In other words, by rotating certain angles, the three points in the two rings can exactly match, and the rotated angles varied from trial to trial, which allowed us to separately decode the location sequence and color sequence in the following analysis. In contrast, the location and color sequences in the misaligned trajectory condition (*MAT* condition) differed both in positions and trajectory distances within rings (*Figure 1C*).

## Aligned color-location trajectory improves memory performance

We first estimated memory precision for color and location sequences by calculating the reciprocal of the circular standard deviation of response error (circular difference between reported location (color) and correct location (color)) across trials ($1/\sigma$) (*Bays et al., 2009*). As shown in *Figure 2A*, two-way repeated ANOVA (alignment (AT vs. MAT)×task (location vs. color)) revealed significant main effects for alignment ($F_{(1,32)} = 4.279$, p=0.047, $\eta_p^2 = 0.118$) and task ($F_{(1,32)} = 139.382$, p<0.001, $\eta_p^2 = 0.813$), but nonsignificant interaction effect ($F_{(1,32)} = 0.618$, p=0.438, $\eta_p^2 = 0.019$). Specifically, the AT condition had better memory performance than the MAT condition, supporting our hypothesis that shared structure facilitates memory of multiple sequences. Moreover, location memory performed better than color memory. Further comparison revealed that the aligned condition mainly enhanced color memory (paired-t test, $t_{(32)} = 2.446$, p=0.020, Cohen's d=0.426) but not location (paired-t test, $t_{(32)} = 1.538$, p=0.134, Cohen's d=0.268). Better location vs. color memory performance indicates that alignment operation is less effective in improving memory (i.e. location sequence) that is already very robust (*Wu et al., 2020*). In terms of serial position in sequence, color sequences demonstrated better memory performance under AT versus MAT conditions, especially for the second and third items (paired-t test, first: $t_{(32)} = -0.315$, p=0.755, Cohen's d=0.055; second: $t_{(32)} = 4.069$, p<0.001, Cohen's d=0.709; third: $t_{(32)} = 2.583$, p=0.015, Cohen's d=0.450) (*Figure 2B*). Meanwhile, the location sequences exhibited similar performance for all positions (paired-t test, first: $t_{(32)} = 0.972$, p=0.338, Cohen's d=0.169; second: $t_{(32)} = 1.245$, p=0.222, Cohen's d=0.216; third: $t_{(32)} = 1.290$, p=0.206, Cohen's d=0.225) (*Figure 2C*). Overall, behavioral findings demonstrate an improvement in WM performance with a common trajectory across feature domains, and indicate that the aligned trajectories (first to second, second to third) may be applied to reduce the memory uncertainty for the second and third colors.

## Aligned color-location trajectory elicits color-location correlation in recalled trajectories

We further investigated whether the location-color trajectory alignment was truly leveraged in the memory process. Note that participants reproduced the color and location sequences by clicking three positions on the respective rings, i.e., reproducing two spatial trajectories, one for location and one for color (*Figure 1A*, reporting period). We, therefore, could examine the correlation between the

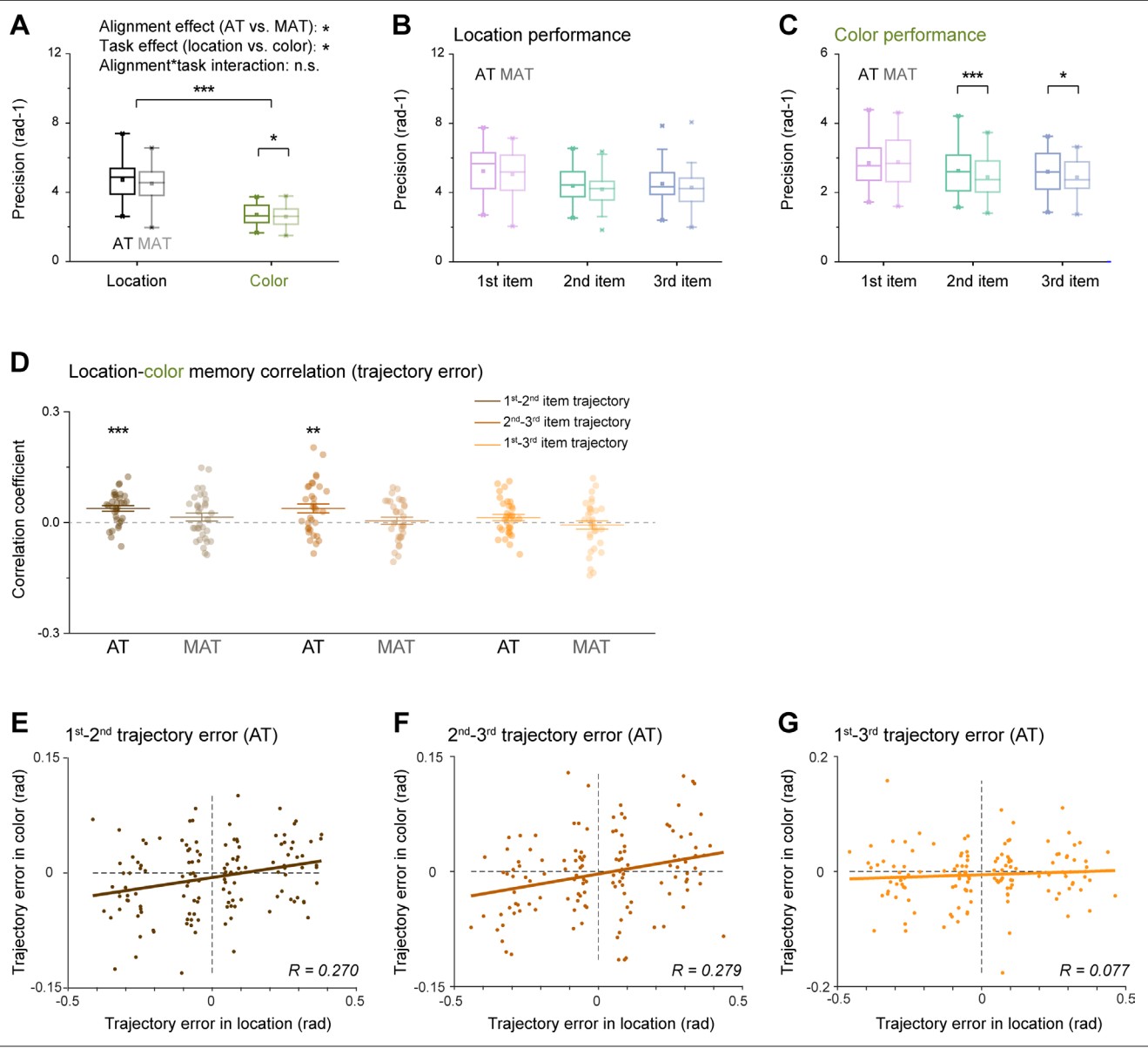

**Figure 2.** Behavioral performance. (**A**) Memory precision performance of location (black) and color (green) sequences for aligned trajectory (AT) (dark color) and misaligned condition (MAT) (light color) conditions. Horizontal line in the boxplots denotes the median; box outlines denote the 25th and 75th percentiles; whiskers denote 1.5 × the interquartile range. Extreme values are denoted by crosses. (N = 33, *p<0.05; **p<0.01; ***p<0.001). (**B**) Memory precision of first (purple), second (turquoise), and third (blue) items of location sequence, for AT (dark color) and MAT (light color) conditions. (**C**) Memory precision of first (purple), second (turquoise), and third (blue) items of color sequence, for AT (dark color) and MAT (light color) conditions. (**D**) Grand average (mean ± SEM) correlation coefficients of recalled trajectory error between location and color sequences, for first-to-second trajectory (brown), second -to-third trajectory (brickred), and first-to-third trajectory (orange), under AT (dark color) and MAT (light color) conditions. Dots indicate individual participants. (**E**) Scatterplot of first-to-second trajectory memory error for location sequence (X-axis) and Color sequence (Y-axis) under AT condition. Note that the trajectory error of all trials within each subject was divided into four bins according to the location trajectory error, resulting in 33 (subject number)*4 (bins) dots in the plot. The brown line represents the best linear fit. (**F**) Same as E, but for the second-to-third trajectory. (**G**) Same as E, but for the first-to-third trajectory.

The online version of this article includes the following figure supplement(s) for figure 2:

**Figure supplement 1.** Behavioral performance and trajectory representation for misaligned condition.

reported location and color trajectories in their maps to determine whether the AT condition would result in a correlated pattern based on the reported sequences.

Specifically, we first calculated trajectory error (the circular difference between the reported trajectory and the true trajectory) for location and color features, and then accessed the correlation between the (signed) trajectory error of location and color features, for each subject. As shown in *Figure 2D*, the AT condition showed significant correlations for both first-second (one-sample t-test, $t_{(32)}=5.022$, p<0.001) and second-third (one-sample t-test, $t_{(32)}=3.113$, p=0.004) trajectories, but not for first- third trajectory (one-sample t-test, $t_{(32)}=1.579$, p=0.124). In contrast, the MAT condition did not display any significant correlation (one-sample t-test; first- second: $t_{(32)}=1.361$, p=0.183; second-third: $t_{(32)}=0.490$, p=0.628; first-third: $t_{(32)}=-0.582$, p=0.565).

At the group level, motivated by a previous study (*Li et al., 2021b*), we quantified the trajectory correlations by first binning all trials based on the location trajectory error, then extracting the color trajectory error for each bin, for each subject and pooling the data across subjects. As shown in *Figure 2* EFG, a significant correlation was observed for the first-second (r=0.270, p=0.002) and second-third trajectory (r=0.279, p=0.002), but not for the first-third trajectory (r=0.077, p=0.375). Similarly, the MAT condition did not exhibit any location-color correlation in trajectories (first-second: r=0.097, p=0.277; second-third: r=0.025, p=0.790; first-third: r=−0.065, p=0.443; also see *Figure 2— figure supplement 1A–C*), which excludes the possibility that the reported trajectory correlation was solely due to systematic response bias. We further confirmed the trajectory errors correlations using a generalized linear mixed-effects model (AT: first-second trajectory, $\beta=0.071$, t=4.215, p<0.001; second-third trajectory, $\beta=0.077$, t=3.570, p<0.001; first-third trajectory, $\beta=0.019$, t=1.118, p=0.264; MAT: first-second trajectory, $\beta=0.031$, t=1.572, p=0.116; second-third trajectory, $\beta=-0.002$, t=0.128, p=0.898; first-third trajectory, $\beta=-0.017$, t=−1.024, p=0.306).

Together, behavioral findings indicate that memory facilitation arises from an automatic alignment of recalled trajectories across feature domains to compress information. In other words, instead of memorizing two three-item sequences independently, subjects may just maintain their common trajectories.

## Neural decoding of location and color features during encoding

We employed a time-resolved inverted encoding model (IEM) (*Brouwer and Heeger, 2009*; *Brouwer and Heeger, 2011*) on EEG signals to examine the neural representation of location and color. Specifically, the slope of the reconstructed channel response was estimated to quantify the time-resolved

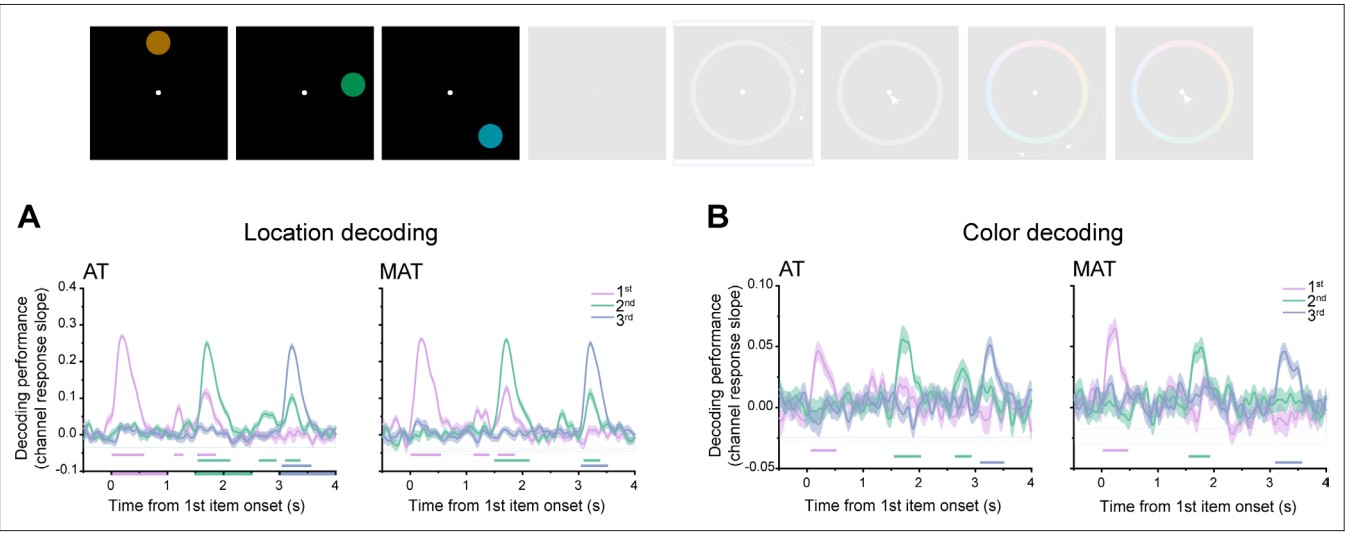

**Figure 3.** Neural representation of memory contents during the encoding period. (**A**) Grand average (mean ± SEM) neural decoding (slope of channel response) of location information for the first (purple), second (turquoise), and third (blue) disk as a function of time during the encoding period, for aligned trajectory (AT) (left panel) and misaligned conditions (MAT) (right panel). Horizontal lines with corresponding colors denote significant time ranges (cluster-based permutation test, cluster-defining threshold p<0.001, corrected significance level p<0.001) (**B**) Same as A, but for color feature decoding.

decoding performance for the first, second, and third location and color, respectively (see details in Materials and methods) at each time point. We first focused on the encoding period when the three-disk sequence was physically presented.

As shown in *Figure 3A*, the location of each of the three disks could be successfully decoded from EEG signals for both AT (first location: 0.03–0.56 s, 1.14–1.26 s, 1.55–1.84 s; second location: 1.56–2.10 s, 2.65–2.92 s, 3.12–3.35 s; third location: 3.06–3.50 s; corrected cluster p<0.001) and MAT conditions (first location: 0.03–0.52 s, 1.15–1.39 s, 1.58–1.84 s,; second location: 1.52–2.10 s, 3.11–3.36 s; third location: 3.06–3.54 s; corrected cluster p<0.001) during stimulus presentation period. Similarly, color information could also be decoded for both AT (first color: 0.09–0.50 s, corrected cluster p<0.001;second color: 1.57–2.01 s, corrected cluster p<0.001; third color: 3.11–3.49 s, corrected cluster p=0.002) and MAT conditions (first color: 0.04–0.45 s; second color: 1.57–1.91 s, corrected cluster p<0.001; third color: 3.11–3.55 s; corrected cluster p<0.001) (*Figure 3B*). It is noteworthy that location and color features were generated with the constraint that they could not occupy the same position within their respective rings. This thereby ensured the independent decoding of location and color features from the same neural signals. Moreover, the color feature exhibited weaker decoding strength than location, also consistent with behavioral results (*Figure 2A*).

## Spontaneous replay of color sequence during location recall

After confirming location and color representations during the encoding period, we next examined the neuronal correlates of sequence memory during retrieval. We are particularly interested in the 'recall location' period, during which subjects need to remain still without making motor responses but at the same time prepare for subsequent location recall (see *Figure 1A*, and upper panel of *Figure 4*). During this period, subjects need to maintain two sequences: the location sequence which is immediately task-relevant, and the color sequence which is not task-relevant right now but will be recalled later. That is to say, we posit that memories should be converted to sequences as outputs during this period. Behavioral analysis indicates the correlation between recalled location and color trajectories for AT condition (*Figure 2*), which suggests an active combination of common trajectories across features. As a result, we sought neural evidence for the reintegration between the color sequence and the location sequence for the AT condition during this period.

As shown in *Figure 4A* (left panel), the currently task-relevant location sequence during AT condition displayed strong decoding performance for the first location (0.11–0.46 s, corrected cluster p=0.003), weak but significant decoding performance for the third location (0.27–0.41 s, corrected cluster p=0.011), but not for the second location. Moreover, The MAT condition (*Figure 4A*, right panel) showed the similar location decoding profiles (first location: 0.11–0.46 s, corrected cluster p<0.001; third location: 0.13–0.35 s, corrected cluster p=0.002). The primacy effect might be due to the fact that the first location is the first to be recalled afterward, and therefore it denotes either the most task-relevant feature or motor preparation. In fact, a similar position effect has also been observed for color sequence during the color recalling period (*Figure 4—figure supplement 1*).

Most importantly, we asked whether the 'recall location' period also contains color sequence information, which is not task-relevant at the moment but will be recalled later. As shown in the left panel of *Figure 4B*, we observed significant reactivation of color sequence for the AT condition. Specifically, the color sequence undergoes a temporally compressed, forward reply (first color: 0.10–0.16 s, corrected cluster p=0.048; second color: 0.21–0.27 s, corrected cluster p=0.046; third color: 0.31–0.38 s, corrected cluster p=0.089). In contrast, the MAT condition did not display any neural reactivation of the color sequence (*Figure 4B*, right panel). Direct comparison between AT and MAT conditions showed a significant reactivation difference (two-way repeated ANOVA, $F_{(1,32)} = 14.213$, p=0.001, $\eta_p^2 = 0.308$). Further analysis (*Figure 4C*) reveals that the AT-MAT difference is mainly due to the first (paired test, $t_{(32)} = 3.151$, p=0.004, Cohen's d=0.548) and second items ($t_{(32)} = 1.914$, p=0.065, Cohen's d=0.334).

We next used a 'sequences' approach (*Kurth-Nelson et al., 2016*; *Liu et al., 2019*) to characterize the sequential replay profile. Specifically, we calculated the cross-correlation coefficients between consecutive items, and then performed a permutation test to examine the statistical significance of the sequential replay profile by shuffling color labels across participants. *Figure 4D* shows a significant temporal lag around 110 ms and 140 ms for the first-second and second-third color pairs, respectively,

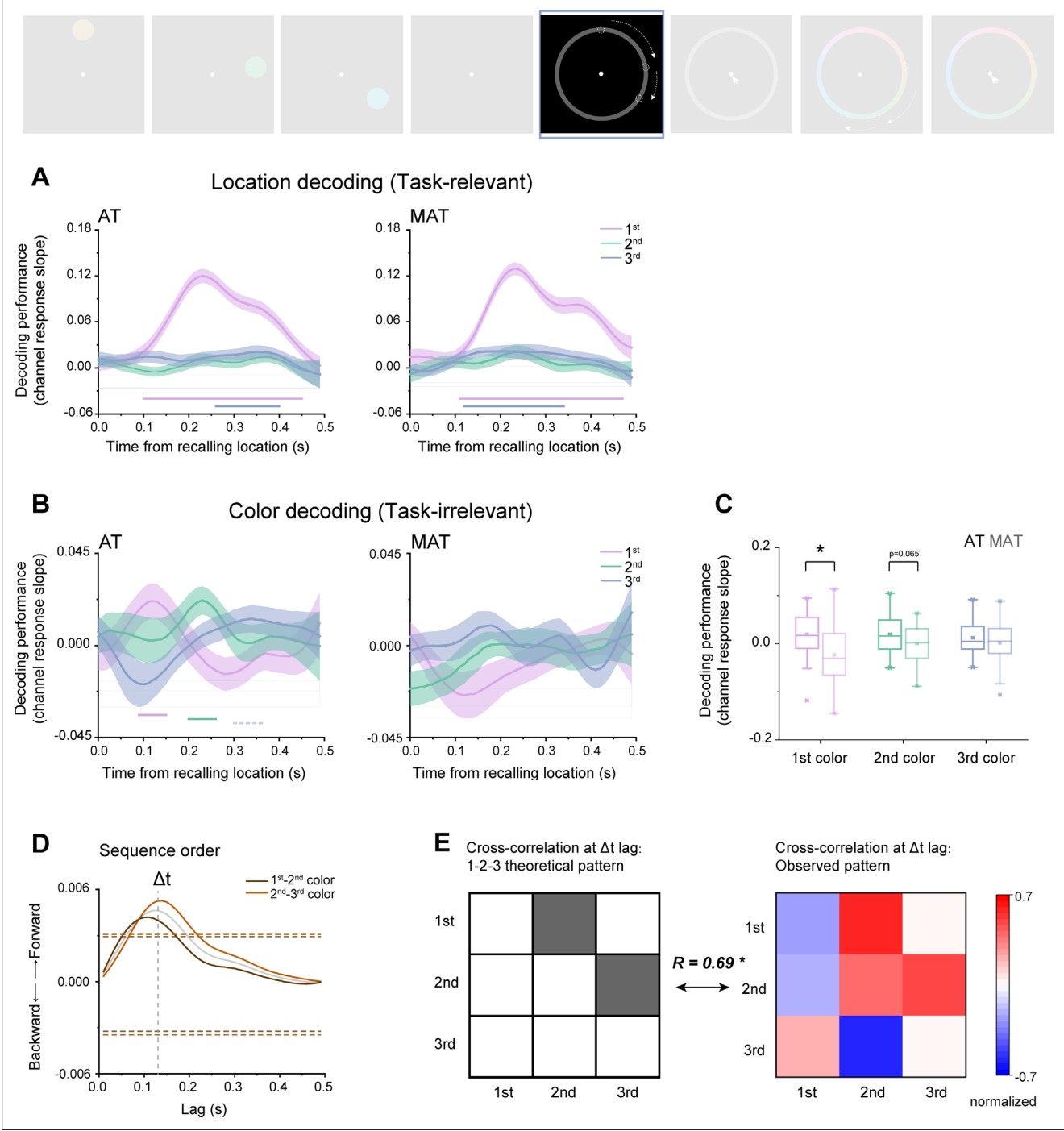

**Figure 4.** Spontaneous color sequence replay during 'location recall'. (**A**) Grand average (mean ± SEM) decoding performance for first (purple), second (turquoise), and third (blue) locations as a function of time during the '"recall location' period, for aligned trajectory (AT) (left panel) and misaligned (MAT) conditions (right panel). (**B**) Grand average (mean ± SEM) decoding performance for first (purple), second (turquoise), and third (blue) colors as a function of time during 'recall location' period, for AT (left panel) and MAT conditions (right panel). (Horizontal solid line: cluster-based permutation test, cluster-defining threshold p<0.05, corrected significance level p<0.05; Horizontal dashed line: marginal significance, cluster-defining threshold p<0.1, 0.05<cluster p<0.1) (**C**) Grand average decoding performance within the respective significant time range, for first (purple), second (turquoise) and third (blue) colors, under AT (dark color) and MAT (light color) conditions. (**D**) Cross-correlation coefficient, calculated to quantify the extent of the neural representations of adjacent two items followed a forward (positive y) or backward (negative y) transition as a funciton of time lag, between first and second colors (brown color) and between second and third (brick red color) colors, and their average (gray color). Dashed vertical line denotes the peak of the averaged cross-correlation time courses. Dashed horizontal lines denote the nonparametric statistical significance threshold (p<0.05, permutation test). (**E**) Left panel: theoretical transition pattern for three-item forward replay, i.e., first-second-third, characterized by cross-correlation at certain time

*Figure 4 continued on next page*

*Figure 4 continued*

lag. Right panel: empirical transitional pattern (actual cross-correlation matrix) at 130 ms time lag. A significant correlation was found between the two matrices (*r*=0.690, p=0.040), further confirming the forward replay of color sequence.

The online version of this article includes the following figure supplement(s) for figure 4:

**Figure supplement 1.** Color and location representations during "color recall".

**Figure supplement 2.** Participants were divided into two groups based on the average of trajectory decoding performance within the respective significant time range.

indicating a forward replay profile with temporal compression within approximately 130 ms (peak of the average of the two cross-correlation time courses).

Moreover, motivated by previous studies (*Liu et al., 2021a*; *Liu et al., 2021b*), we first constructed the theoretical transitional pattern for the three-item sequence by assuming a Δt temporal lag between consecutive items (i.e. cross-correlation matrix). As shown in *Figure 4E* (left panel), the first item at time T could predict the reactivation of the second item at T+Δt, and the second item at time T could predict the appearance of the third item at T+Δt. We then calculated the actual cross-correlation matrix (empirical transitional pattern) at 130 ms time lag which denotes the time lag of consecutive items in our findings (see *Figure 4D*, gray line), resulting in a 3×3 matrix (*Figure 4E*, right panel). A significant correlation was found between the empirical cross-correlation matrix and the theoretical transitional pattern (*r*=0.690, p=0.040), further confirming the forward replay of the color sequence.

Together, when subjects prepare to reproduce location sequence during 'recall location,' the currently task-irrelevant color sequence demonstrates a spontaneous sequential replay profile, which highlights the close bond between color and location sequences when they share a common trajectory. Together with the color-location trajectory correlation in behavior, the findings suggest that replay-based neural mechanisms in WM mediate sequence combinations based on common structures.

## Neural representation of common trajectory and its behavioral correlates

Finally, we accessed the neural representations of the common trajectory structure during encoding and retention periods. Specifically, a linear support vector machine (SVM) was employed to decode the first-to-second and second-to-third trajectory distance, with the first-to-third trajectory as a control. Since there are only eight possible circular distances between every two locations on a ring, the chance level of decoding performance is 0.125.

During the encoding period (*Figure 5A*), the first-to-second trajectory appeared right after the presentation of the second item for AT condition (1.55–2.11 s, corrected cluster p<0.001), which is expected since the relationship between the first and second items along the trajectory can only be established when the second item occurs. Similarly, the second-to-third trajectory appeared after onset of the third item (3.10–3.48 s, corrected cluster p<0.001) (*Figure 5B*). In contrast, the first-third trajectory showed nonsignificant neural decoding (*Figure 5C*), consistent with the nonsignificant color-location trajectory correlation in behavior. Most interestingly, the first-to-second trajectory was reactivated right after the third item (3.19–3.48 s, corrected cluster p=0.006; *Figure 5A*), implicating the formed link between the two trajectories (first-second and second-third) for aligned location-color sequences. Therefore, when the color and location sequences share the same trajectory (i.e. AT condition), brain activities tend to co-represent the previously formed first-to-second trajectory (reactivation) and the newly formed second-to-third trajectory during encoding to establish the full trajectory (see *Figure 2—figure supplement 1D, E, F* for MAT condition). The trajectory reactivation was also related to WM behavior. Specifically, we divided all participants into two groups based on their first-to-second color-location trajectory memory correlations in behavior (*Figure 2D*) and calculated their corresponding neural representation of the first-to-second trajectory, respectively. Both the higher- and lower-correlation groups displayed significant neural decoding of the first-to-second trajectory right after the second item (*Figure 5D*; higher group: 1.56–1.99 s, corrected cluster p<0.001; lower group: 1.64–1.97 s, corrected cluster p<0.001). Meanwhile, only the higher-correlation group exhibited a significant reactivation of the first-to-second trajectory after the onset of the third item (3.25–3.46 s, corrected cluster p=0.018, 3.55–3.68 s, corrected cluster p=0.088; Higher group vs. Lower group: bootstrap test, p=0.037, two-side). Moreover, we were curious about the correlation

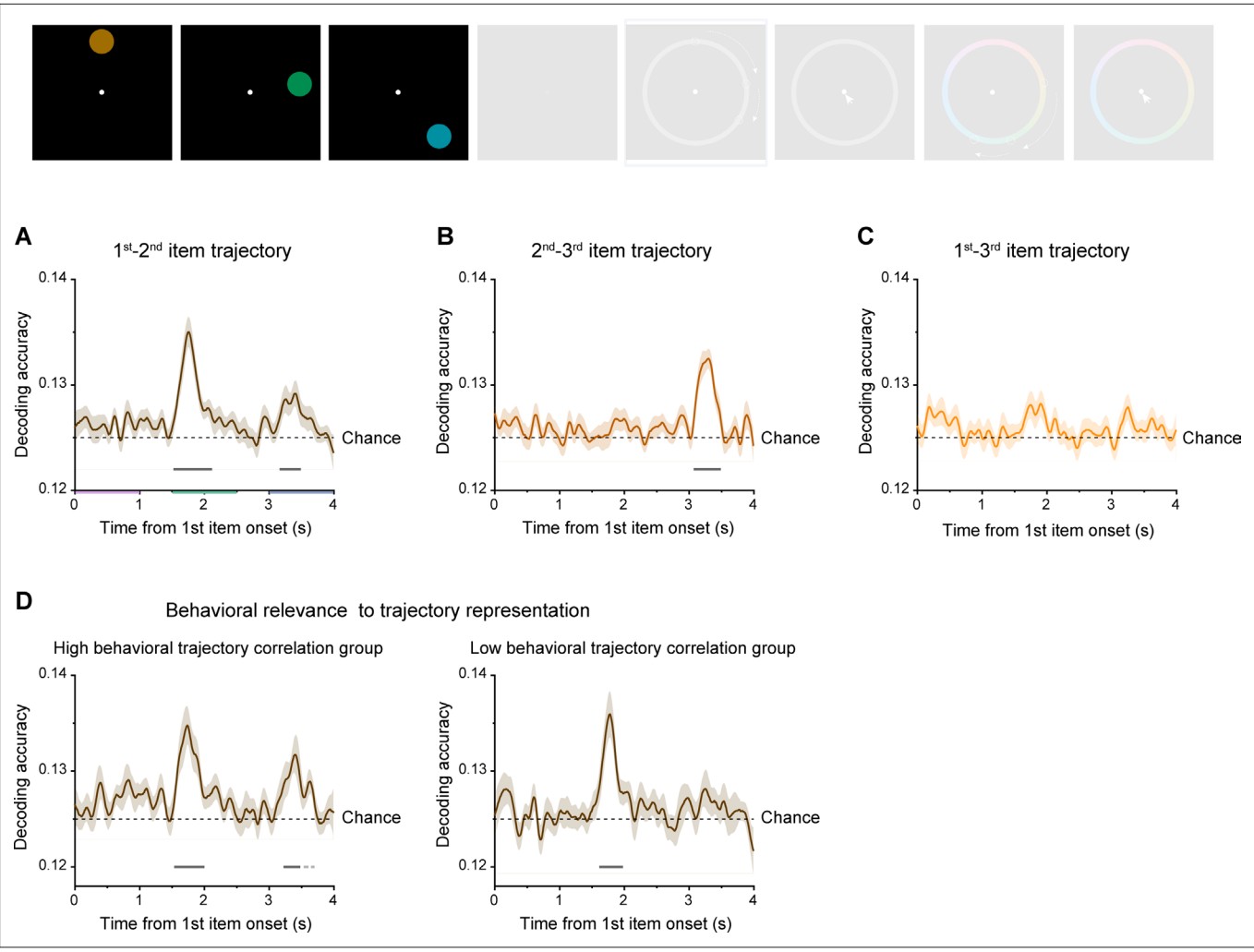

**Figure 5.** Common trajectory representation and its behavioral relevance. (**A**) Grand average (mean ± SEM) neural decoding of first-to-second as a function of time during the encoding period, for aligned trajectory (AT) condition. (**B**) Same as A, but for second-to-third trajectory. (**C**) Same as A, but for first-to-third trajectory. (**D**) Participants were divided into two groups (higher-correlation group and lower-correlation group), based on their first-to-second color-location trajectory memory behavioral correlation. Grand average (mean ± SEM) neural decoding of first-to-second trajectory as a function of time, for higher-correlation group (n=16; left panel) and lower-correlation group (n=16; right panel). (Horizontal solid line: cluster-based permutation test, cluster-defining threshold p<0.05, corrected significance level p<0.05; Horizontal dashed line: cluster-based permutation test, cluster-defining threshold p<0.05, corrected significance level p<0.1).

The online version of this article includes the following figure supplement(s) for figure 5:

**Figure supplement 1.** Trajectory representation during memory maintaining period.

between the common trajectory representation and latter forward replay pattern. Therefore, based on the average of trajectory neural representation (first-second and second-third trajectories) during presentation of the third item when common trajectory reactivation was observed, we divided all participants into two groups, and observed higher trajectory representation group was accompanied with clearer forward replay (see *Figure 4—figure supplement 2*). This finding indicates that memory reorganization formed during encoding can predict serial replay during recalling.

With regards to the retention period, based on previous literatures (*de Vries et al., 2020*; *Foster et al., 2016*; *Fukuda et al., 2016*; *Sutterer et al., 2019*), we chose to rely on the alpha-band activities to decode shared trajectories (see decoding results based on raw signals in *Figure 5— figure supplement 1A, B*). The aligned condition showed significant and long-lasting decoding of compression trajectories (*Figure 5—figure supplement 1C, D*), while the misaligned condition only showed decoding at the beginning, which might be due to the non- offset response of the third item (*Figure 5—figure supplement 1G, H*). The results, although not as clear as those during encoding

and recalling periods, also supports compression trajectory is spontaneously leveraged to reorganize multiple information.

## Discussion

Identifying the underlying structure to facilitate efficient information storage in WM is crucial to human intelligence. Here, we investigated whether common structures shared across different feature domains would be spontaneously employed to facilitate memory of multiple sequences and its neural correlates. Since both location and color features could be characterized by positions along a continuous ring, we systematically manipulated the trajectory consistency between the location and color sequences. We show that color-location trajectory alignment is associated with better memory performance than the misaligned condition, and the memory benefit is attributed to structure-induced constraints on individual items that decrease representational uncertainty. EEG recordings provide neural evidence for the employed 'compressive' strategy, i.e., reactivation of shared trajectories during encoding and retention, and spontaneous replay of color sequences during location recall. Finally, structure reactivation is related to WM behavior and neural replay. Together, shared common structure is leveraged for storage of multiple sequences through compressed encoding and neural replay, together facilitating efficient information organization in WM.

Events in daily experiences are not isolated but are always linked to each other. Therefore, instead of treating individual events as independent information, a more efficient way is to seek the link between seemingly unrelated events, i.e., 'connecting the dots' in the WM system. Here the trajectories for both location and color sequences are defined in a ring coordinate system, providing an abstract-level cognitive map for memory formation. Our results demonstrate that subjects spontaneously realign sequence trajectories across features to facilitate memory of two sequences. In other words, instead of memorizing two three-item sequences, subjects could just maintain two starting points and a common trajectory, an apparently more efficient way. Not that the trajectory alignment manipulation differs from the Gestalt principles of perceptual organization such as proximity and similarity principles (*Goldstone and Medin, 1994*), and it instead reflects a higher-order relationship between maps. The alignment manipulation also could not be accounted for by associative memory (*Aho et al., 2022*; *Roads and Love, 2020*), since the rotational orientation for alignments between the two maps differed on a trial-by-trial basis. Finally, our study is also different from recent works on structure learning and generalization (*Dekker et al., 2022*; *Garvert et al., 2017*; *Liu et al., 2021b*; *Ren et al., 2022*; *Schapiro et al., 2013*), as our task does not involve pre-exposure training or task-related rewards.

Notably, our findings could also be understood in terms of schematic abstraction, which plays pivotal roles in memory formation (*Gilboa and Marlatte, 2017*; *Tse et al., 2007*), such that a congruent schema would facilitate memory retrieval compared to an incongruent condition (*Audrain and McAndrews, 2022*; *van Kesteren et al., 2010*). In fact, cognitive map refers to the internal representation of spatial relations in a specific environment (*Tolman, 1948*), while schematic abstraction denotes a more broad range of circumstances, whereby the gist or structure of multiple environments or episodes can be integrated (*Bartlett, 1932*; *Farzanfar et al., 2023*). In other words, schema refers to a highly abstract framework of prior knowledge that captures common patterns across related experiences, which does not necessarily occur in a spatial framework as cognitive maps do. In the current design, as we specifically manipulate the consistency of spatial trajectory distance between color and location sequences defined in a spatial map, cognitive map might be a more conservative term to frame our findings, although in essence it reflects general schema-based WM reorganization.

The fact that without task requirement, human participants still spontaneously extracted underlying common structure and leveraged it to organize multiple item storage reflects the intelligence of our brain to achieve efficient information coding (*Attneave, 1954*). Indeed, there is a mountain of research suggesting that participants exploit statistical regularities to form efficient WM representation without explicit instructions to do so (e.g. *Brady et al., 2009*; *Brady and Tenenbaum, 2013*). However, it remains largely unknown about the underlying neural mechanism. Here, the common structure we manipulated is inspired by the theory of cognitive map (*Behrens et al., 2018*; *Bellmund et al., 2018*; *O'keefe and Nadel, 1978*), which has argued that reasoning in abstract domains follows similar computational principles as in spatial domains. This theory has been supported by accumulated neuroscientific evidence suggesting common neural substrates for knowledge representation

across domains (*Constantinescu et al., 2016*; *Garvert et al., 2017*; *Park et al., 2021*; *Schuck et al., 2016*; *Solomon et al., 2019*; *Theves et al., 2019*). In line with these evidence, recent behavioral studies further prove the integration of information representation across domains based on common computing principles. For example, learning process is accelerated when two different feature maps are aligned (*Aho et al., 2022*), and distance-dependent generalization is observed across two different domains in order to search for correlated rewards (*Wu et al., 2020*). Here, we extend the functional role of cognitive map in efficiently organizing information across domains in human WM and reveal compressive encoding and neural replay in facilitating multiple sequence storage.

Neural replay refers to the sequential reactivation in the same or reversed order as previous experience. It was first observed in the rodent hippocampus and mainly for spatial navigation (*Wilson and McNaughton, 1994*), but has recently been found in many higher-level non-spatial tasks in human brains (*Kurth-Nelson et al., 2016*; *Liu et al., 2019*; *Liu et al., 2021b*; *Schapiro et al., 2018*; *Schuck and Niv, 2019*; *Zhang et al., 2018*). In fact, neural replay has been posited to represent abstract structure (*Huang et al., 2021*; *Liu et al., 2019*), structure-based inference (*Liu et al., 2021b*), and generalization (*Barry and Love, 2022*). Here, when subjects prepare to recall location sequences ('location recall'), neural replay occurs for color sequences but not for location sequences, supporting the spontaneous nature of neural replay, since color features are not task-relevant right now. The results also exclude other interpretations, such as motor preparation, eye movements, attentional sampling, sequential rehearsal, etc., since if that is the case, we would expect a similar neural replay profile for location sequences that are to be serially recalled soon. Furthermore, color neural replay only appears for the color-location aligned sequence (AT condition) but not for the misaligned sequences (MAT condition), implicating that neural replay serves to consolidate sequences that share a common structure. Therefore, our findings demonstrate new roles of neural replay in structure representation, that is, mediating structure alignment between sequences in the WM system.

It is posited that structure and content are represented in a factorized manner (*Behrens et al., 2018*; *Bengio et al., 2013*), and sequence structure representation that is independent of attached contents guides the replay of new experiences (*Liu et al., 2019*). Factorization representation is thought to help fast generalization of a previously learned structure to new contents (*Sheahan et al., 2021*; *Zhou et al., 2021*). Indeed, the ability to spontaneously perceive relational structures is posited to signify the major distinction between human and nonhuman primates (*Dehaene et al., 2015*; *Zhang et al., 2022*). Meanwhile, previous modeling works also suggest that higher-order structures incorporated in WM would serve as constraints on individual-item representations to reduce representational uncertainty (*Brady and Tenenbaum, 2013*; *Ding et al., 2017*). In this work, we provide behavioral and neural evidence that structure is not only dissociated from content representation, i.e., factorization coding, but also can be aligned in a spontaneous manner, i.e., linking structures, which together contribute to efficient representation of memory information.

## Materials and methods

### Participants

Thirty-six participants (18 males, age ranging from 17 to 25 years) were recruited to accomplish our multi-sequence working memory task. Three participants were removed, since they could not finish the whole experiment. No statistical methods were used to predetermine sample sizes, but our sample sizes are similar to previous studies (*Li et al., 2021a*; *Wolff et al., 2017*). All participants had normal or corrected-to-normal vision with no history of neurological disorders. They were naïve to the purpose of the experiments, and provided written informed consent prior to the start of the experiment. This study received ethical approval from the Peking University Research Ethics Committee (reference number 2020-03-03). This study was carried out in accordance with the Declaration of Helsinki.

Participants sat in a dark room, 60 cm in front of a Display ++ monitor with 100 Hz refresh rate and a resolution of 1920 × 1080, and their head stabilized on a chin rest. At the beginning of trial, three disks (1.5° × 1.5° visual angle) were sequentially presented at different locations of the screen, with different colors. The spatial location of each disk was independently drawn from a fixed set of 9 locations, which were evenly distributed on an imaginary circle with radius of 7° visual angle from central fixation and spaced 40° from the nearest locations, with a small random jitter (±1° – ±3°) added to each. The color of the each disk was also independently selected from a fixed set of nine colors, which were evenly

distributed along a circle in Commission Internationale de l'Eclairage (CIE) L*a*b* space, and equidistant from the gray point at L*=50, a*=0, and b*=0 (*Brouwer and Heeger, 2009*), and spaced by 40°, with a small random jitter (±1° – ±3°). Each disk was presented for 1 s, with 0.5 s interval between two adjacent disks. After 2 s delay during which only the fixation point remained on screen, a gray ring appeared for 0.5 s with the same radius (7° visual angle) from central fixation to instruct participants to recall three spatial locations without any movement. Then, a cursor appeared at fixation, and participants should report the remembered locations sequentially in their presented order by using a mouse to click on the gray location ring. After delivering three spatial location responses, a color ring was presented for 0.5 s (7° visual angle in radius) to instruct participants to recall three colors without any movement. Similarly, a cursor then appeared, and participants were asked to report the remembered colors sequentially in their presented order by clicking on the color ring. To reduce the complexity of the task, the color wheel was oriented the same way for individual participant.

Note that even though color and location were different features, their values were both chosen from nine positions/values based on their respective ring (0° – 320° in 40° increments), with the constraint that the color value and location value for the same item can't be the same. In order to investigate whether common structure would organize multiple information storage in different domains, we modulated trajectory consistency. Specifically, in the aligned trajectory condition (AT), both the first-to-second and second-to-third disk trajectory distances in location domain were the same as that in color domain. In other words, by rotating certain degree, the whole trajectory (from the first to third point) was matched in the location and color maps. At the same time, we varied the rotated degree to align the two maps on a trial-by-trial basis, such that we could not predict color sequence solely based on location sequence. This manipulation was critical to independently decode color sequence and location sequence. In misaligned trajectory condition (MAT), the whole trajectories (first-second-third) in the two maps were different, while partial trajectory (either first-second or second-third) can be the same for some trials. Therefore, in MAT condition, we couldn't rotate one map to exactly match the other map. Note that random selection for individual item both for AT and MAT conditions resulted in varied trajectories, which can move in clockwise or anticlockwise direction and the direction can even be reversed on the third item. Trials from AT and MAT conditions were interleaved, aiming to investigate the spontaneous information organization process in a more natural way and avoid prediction about trial type. In each trial three locations were chosen independently from 9 values (0° – 320° in 40° increments, each occurred 36 times with random order), but with a constraint that they should at least differ by 40°. The same rule was applied to three colors. Moreover, the color and location value from the same object were also constrained to be different. Participants should complete 648 trials in total, which was divided into two sessions on two separate days, separated by at most one week. It took approximately 3 hr to accomplish one session (including breaks).

## EEG acquisition and preprocessing

The EEG data was recorded using a 64-channel EasyCap and two BrainAmp amplifiers (BrainProducts). Horizontal electrooculography (EOG) was recorded by an additional electrode around the participants' right eye. The impedances of all electrodes were kept below 10 k. The EEG data was preprocessed offline using FieldTrip software (*Oostenveld et al., 2011*). Specifically, the data was first referenced to the average value of all channels, band-pass filtered between 2 and 50 Hz, and down-sampled to 100 Hz. The data was then baseline-corrected, by selecting the time range from 300 ms to 100 ms before the presentation of the first disk in each trial as baseline to be subtracted. Then, independent component analysis (ICA) was performed independently for each participant to remove eye movement and artifact components, and the remaining components were back-projected onto the EEG electrode space. To further identify artifacts, we calculated the variance (collapsed over channels and time) for each trial. Trials with excessive variances were removed. Note that the following decoding approach was based on the whole electrodes, except that location decoding in the encoding period was based on the posterior electrodes (P7, P5, P3, P1, Pz, P4, P6, P8, PO7, PO3, POz, PO4, PO8, O1, Oz, and O2), considering eye movement was not strictly controlled in the present study.

## Data analysis

### Behavioral performance analysis

For each spatial location and color, the response error was first quantified by the circular difference between the reported location (color) and the true target location (color) in each trial. The memory precision was then estimated by calculating the reciprocal of circular standard deviation of response error. To explore the similarity of the perceived trajectory in spatial location and color domains, we calculated the circular correlation of the perceived trajectory (trajectory memory error) between the two domains for each participant. Trajectory response error was quantified by the circular difference between the reported trajectory and the true trajectory, e.g., the first-to-second trajectory error in location was calculated by the difference between the first location error and second location error. In group level, we quantified the circular correlations of trajectory error by first sorting trials into four bins based on their location trajectory error for each participant, then binning the trials, computing the color trajectory error for each bin, and pooling the data across participants.

### Time-resolved location and color decoding

Similar as previous studies (*Brouwer and Heeger, 2009*; *Brouwer and Heeger, 2011*; *Huang et al., 2021*), in order to assess the time-resolved location and color information from the EEG signals, we implemented the inverted encoding model (IEM) to reconstruct the location and color information from the neural activities at each time point. The IEM assumes that the response in each sensor could be approximated as a linear sum of underlying neural populations encoding different values of the feature-of-interest (i.e. tuning channels). Here, the number of location and color tuning channels were both set to 9. Following previous work (*Ester et al., 2015*; *Yu et al., 2020*), the idealized feature tuning curves of nine channels were defined as nine half-wave rectified sinusoids centered at different location (color) values (0°, 40°, 80°, and so on) and raised to the eighth power.

We began by modeling the response of each EEG sensor as a linear sum of nine information channels, characterized by $B_1 = WC_1$, in the training data set, where $B_1$ (m sensors × n trials) represents the observed response at each sensor, $C_1$ (k channels × n trials) represents the predicted channel responses, $W$ (m sensors × k channels) represents the weight matrix that characterizes the linear mapping from 'channel space' to 'sensor space.' Therefore, given $B_1$, and $C_1$, the weight matrix $W$ (m sensors × k channels) was calculated by using least-squares regression $\widehat{W} = B_1 C_1^T \left( C_1 C_1^T \right)^{-1}$. Finally, the channel responses ($C_2$) for the test data set ($B_2$) could be extracted using the estimated $\widehat{W}$, by $\hat{C}_2 = \left( \widehat{W}^T \widehat{W} \right)^{-1} \widehat{W}^T B_2$.

Regarding the division of training and test set, a leave one-out cross-validation was implemented, such that data from all but one block was acted as $B_1$ to estimate $W$, while data from the remaining block was acted as $B_2$ to estimate $C_2$. This procedure ensures the independence between training set and testing set. The entire analysis was repeated until all blocks could be held out as a test set. The observed channel responses $C_2$ were then circularly shifted to a common center (0°) in reference to the location/color-of-interest in each trial, and averaged across trials for further analysis.

Consistent with previous studies (*Foster et al., 2017*; *Huang et al., 2021*), decoding performance was characterized by the slope of the estimated channel responses at each time by flipping the reconstructed curves across the center, averaging both sides, and performing linear regression. We further smoothed the slope time courses with a Gaussian kernel (s.d.=40 ms) (*Huang et al., 2021*; *Wolff et al., 2017*).

Note that considering the color wheel is always oriented the same way for each participant, it is possible that participants tend to map colors to locations and remembered in a location code. However, this possibility shouldn't affect the interpretation of the comparison between AT and MAT conditions. In fact, the underlying logic of the current design is based on the facts that thinking spatially is intuitive and color and location sequences can be spontaneously combined or integrated based on the shared common trajectory structure. In other words, decoding of color sequences could be understood as neural representation of a series of corresponding locations along the ring that are independent of the physical locations of the items.

## Forward sequence measure

Following previous studies (*Huang et al., 2018*; *Kurth-Nelson et al., 2016*; *Liu et al., 2019*), cross-correlation was applied to examine whether the color reactivation pattern tended to follow certain order, e.g., a forward (first-secondthird) or reverse order (third-second -first). If it was a forward sequence, the decoded performance of the first item at time T should be correlated with the decoding performance of second item at time T + Δt, and correlated with the decoding performance of third item at time T+2*Δt, where Δt defines a lag between neural representations of two consecutive items. We first calculated the cross-correlation between the first and second items and between second and third items at each time lag. Then we subtracted the reverse direction (second-first; 3$^{rd}$-second) from the forward direction (first-second; second-third) respectively at each time-lag, in order to exclude the autocorrelation effect (*Kurth-Nelson et al., 2016*). The resulting cross-correlation time courses were then averaged to determine the time lag for two consecutive items, here, the time point of the peak of the averaged cross-correlation time course. As mentioned above, for a forward replay pattern, at time lag Δt, we would expect to observe significant transition from the first to second items, and from the second to third items, while nonsignificant transition/correlation for the rest pairs, characterized by a theoretical forward transition pattern in *Figure 4E* (left panel). Meanwhile, the actual cross-correlation matrix can be estimated by computing the correlation coefficients for every pair (first-firstfirst-second, first-third; second-first, second-second, second-third; third-first, third-second, third-third) at the defined time lag (*Figure 4D*). Finally, we quantified the similarity between the observed transition pattern and theoretical forward transition pattern.

## Time-resolved trajectory decoding

We implemented a linear support vector machine (SVM) to decode trajectory distance. Considering there were eight possible distances between every two items, i.e., ±160°, ±120°, ±80°, ±40° (chance level is 1/8), an eight-way decoder (One-VS-rest multiclass classifier) was used to decode trajectory. A fivefold cross-validation scheme was used, and the classification accuracy was averaged across the folds. We repeated this process 50 times with each containing a new random partition of data into fivefolds, and then computed their mean accuracy. Note that color and location sequences shared the same trajectory distances in AT condition, while MAT condition involved different trajectories for location and color. The trajectory decoding in MAT condition was based on the trajectory distance in location domain, considering location information showed much stronger representation than color information (*Figure 2*). Same decoding approach was then employed on alpha band power during maintaining period. Here, alpha-band (8–12 Hz) power time courses was extracted by conducting a time-frequency analysis using Morlet wavelets with a width of seven cycles (Fieldtrip toolbox).

## Statistical analysis

To determine statistical significance of decoding performance time courses, we performed cluster-based permutation test (FieldTrip, cluster-based permutation test, 1000 permutations) (*Maris and Oostenveld, 2007*). We first identified clusters of contiguous significant time points (p<0.05 or p<0.001(during encoding period), two-tailed) from the calculated statistics (one-sample t-test, against 0 (slope value of the reconstructed channel response) for location/color decoding, or against 0.125 (classifier chance level) for trajectory distance decoding), and cluster-level statistics was calculated by computing the size of the clusters. Next, a Monte Carlo randomization procedure was conducted to estimate the significance probabilities for each cluster. Specifically, 0 (for location/color decoding) or 0.125 (for trajectory decoding) with the same sample size was generated and shuffled with the original data 1000 times, and the cluster-level statistics were then calculated from the surrogate data to estimate the significance probabilities for each original cluster.

To determine statistical significance of cross-correlation coefficient (forward direction minus reverse direction), we performed a permutation test by shuffling color labels across participants 1000 times and followed the same procedure to calculate the cross-correlation time courses of the surrogate data, from which the 0.05 threshold level was estimated.

## Acknowledgements

This work was supported by the National Science and Technology Innovation STI2030-Major Project (2021ZD0204103 to HL), National Natural Science Foundation of China (31930052 to HL), and Humboldt Research Fellowship for Postdocs to QH. We thank Christian F Doeller and Muzhi Wang for their helpful comments.

## Additional information

### Competing interests

Huan Luo: Reviewing editor, *eLife*. The other author declares that no competing interests exist.

### Funding

| Funder | Grant reference number | Author |
|---|---|---|
| National Science and Technology Innovation STI2030-Major Project | 2021ZD0204103 | Huan Luo |
| National Natural Science Foundation of China | 31930052 | Huan Luo |
| Humboldt Foundation | Humboldt Research Fellowship | Qiaoli Huang |

The funders had no role in study design, data collection and interpretation, or the decision to submit the work for publication. Open access funding provided by Max Planck Society.

### Author contributions

Qiaoli Huang, Conceptualization, Resources, Data curation, Software, Formal analysis, Validation, Investigation, Visualization, Methodology, Writing - original draft, Project administration, Writing - review and editing; Huan Luo, Conceptualization, Resources, Supervision, Funding acquisition, Writing - original draft, Project administration, Writing - review and editing

### Author ORCIDs

Qiaoli Huang http://orcid.org/0000-0003-4592-9270
Huan Luo http://orcid.org/0000-0002-8349-9796

### Ethics

This study received ethical approval from the Peking University Research Ethics Committee (reference number 2020-03-03). This study was carried out in accordance with the Declaration of Helsinki. All participants provided written informed consent prior to the start of the experiment.

Reviewer #1 (Public Review): https://doi.org/10.7554/eLife.93158.3.sa1
Author response https://doi.org/10.7554/eLife.93158.3.sa2

## Additional files

### Supplementary files
• MDAR checklist

### Data availability

Behavioral data, EEG decoding results and associated code have been deposited at the Open Science Framework (https://osf.io/pswxu/).

The following dataset was generated:

| Author(s) | Year | Dataset title | Dataset URL | Database and Identifier |
|---|---|---|---|---|
| Huang Q, Luo H | 2024 | Shared structure facilitates working memory of multiple sequences | https://osf.io/pswxu/ | Open Science Framework, pswxu |

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
