## [Editor Report · eLife assessment]

This **valuable** study uses a novel experimental design to elegantly demonstrate how we exploit stimulus structure to overcome working memory capacity limits. The presented behavioural and neural evidence are **solid** and in line with the proposed information compression mechanism. This study will be of interest to cognitive neuroscientists studying structure learning and memory.

---

## [Referee Report · Reviewer #1 (Public Review)]

Summary:

Huang and Luo investigated whether regularities between stimulus features can be exploited to facilitate the encoding of each set of stimuli in visual working memory, improving performance. They recorded both behavioural and neural (EEG) data from human participants during a sequential delayed response task involving three items with two properties: location and colour. In the key condition ('aligned trajectory'), the distance between locations of successively presented stimuli was identical to their 'distance' in colour space, permitting a compression strategy of encoding only the location and colour of the first stimulus and the relative distance of the second and third stimulus (as opposed to remembering 3 locations and 3 colours, this would only require remembering 1 location, 1 colour, and 2 distances). Participants recalled the location and colour of each item after a delay.

Consistent with the compression account, participants' location and colour recall errors were correlated and overall lower compared to a non-compressible condition ('misaligned trajectory'). Multivariate analysis of the neural data permitted decoding of the locations and colours during encoding. Crucially, the relative distance could also be decoded - a necessary ingredient for the compression strategy.

Strengths:

The main strength of this study is a novel experimental design that elegantly demonstrates how we exploit stimulus structure to overcome working memory capacity limits. The behavioural results are robust and support the main hypothesis of compressed encoding across a number of analyses. The simple and well-controlled design is suited to neuroimaging studies and paves the way for investigating the neural basis of how environmental structure is detected and represented in memory. Prior studies on this topic have primarily studied behaviour only (e.g., Brady & Tenenbaum, 2013).

Weaknesses:

The main weakness of the study is that the EEG results could make a clearer case for compression. There is some evidence that distance decoding is present in alpha-band activity in the maintenance delay, but the strongest evidence for this occurs only briefly in the late encoding phase (the re-activation of decoding of the distance between items 1 and 2, Fig. 5A). The link to behaviour (Fig. 5D) seems fairly weak and based on a potentially circular analysis. During location recall, colour decoding re-emerges and is reactivated in sequence, but this finding is consistent both with compression-based and conventional rehearsal mechanisms. Nevertheless, the balance of evidence appears to favour the compression account.

Impact:

This important study elegantly demonstrates that the use of shared structure can improve capacity-limited visual working memory. The paradigm and approach explicitly link this field to recent findings on the role of replay in structure learning and will therefore be of interest to neuroscientists studying both topics.

---

## [Author Response]

The following is the authors’ response to the original reviews.

**eLife assessment**
This valuable study uses a novel experimental design to elegantly demonstrate how we exploit stimulus structure to overcome working memory capacity limits. While the behavioural evidence is convincing, the neural evidence is incomplete, as it only provides partial support for the proposed information compression mechanism. This study will be of interest to cognitive neuroscientists studying structure learning and memory.
**Public Reviews:**

**Reviewer #1 (Public Review):**
Summary:Huang and Luo investigated whether regularities between stimulus features can be exploited to facilitate the encoding of each set of stimuli in visual working memory, improving performance. They recorded both behavioural and neural (EEG) data from human participants during a sequential delayed response task involving three items with two properties: location and colour. In the key condition ('aligned trajectory'), the distance between locations of successively presented stimuli was identical to their 'distance' in colour space, permitting a compression strategy of encoding only the location and colour of the first stimulus and the relative distance of the second and third stimulus (as opposed to remembering 3 locations and 3 colours, this would only require remembering 1 location, 1 colour, and 2 distances). Participants recalled the location and colour of each item after a delay.Consistent with the compression account, participants' location and colour recall errors were correlated and were overall lower compared to a non-compressible condition ('misaligned trajectory'). Multivariate analysis of the neural data permitted decoding of the locations and colours during encoding. Crucially, the relative distance could also be decoded - a necessary ingredient for the compression strategy.Strengths:The main strength of this study is a novel experimental design that elegantly demonstrates how we exploit stimulus structure to overcome working memory capacity limits. The behavioural results are robust and support the main hypothesis of compressed encoding across a number of analyses. The simple and well-controlled design is suited to neuroimaging studies and paves the way for investigating the neural basis of how environmental structure is detected and represented in memory. Prior studies on this topic have primarily studied behaviour only (e.g., Brady & Tenenbaum, 2013).

Thanks for the positive comments and excellent summary.

Weaknesses:The main weakness of the study is that the EEG results do not make a clear case for compression or demonstrate its neural basis. If the main aim of this strategy is to improve memory maintenance, it seems that it should be employed during the encoding phase. From then on, the neural representation in memory should be in the compressed format. The only positive evidence for this occurs in the late encoding phase (the re-activation of decoding of the distance between items 1 and 2, Fig. 5A), but the link to behaviour seems fairly weak (p=0.068).

Thanks for raising this important concern. The reviewer is correct that in principle subjects should employ the compression strategy during the encoding phase when sequence stimuli are presented, yet our results show that the 1-2 trajectory could only be decoded during the late encoding phase.

Meanwhile, subjects could not get enough information to form the compressed strategy for the location and color sequences until the appearance of the 3rd item. Specifically, based on the first two items, the 1st and 2nd item, they only learn whether the 1st-2nd trajectories are congruent between location and color features. However, they could not predict whether it would also apply to the incoming 2nd-3rd trajectory. This is exactly what we found in neural decoding results. The 1st-2nd trajectory could be decoded after the 2nd item presentation, and the 2nd-3rd trajectory appears after the 3rd item onset. Most critically, the 1st-2nd trajectory is reactivated after the 3rd item but only for alignment condition, implicating formation of the full-sequence compression strategy wherein the previously formed 1st-2nd trajectory is reactivated to be connected to the 2nd-3rd trajectory.

Regarding the difference between higher- and lower-correlation groups, previously we used the time window based on the overall 2nd-3rd neural reactivations, which might not be sensitive to reactivation strength. We now re-chose the time window based on the higher-correlation group (bootstrap test, p = 0.037, two sides).

Results have been updated (Figure 5; Results, Page 16). Interpretations about the formation of compression strategy during encoding phase have been added to Results (Page 15-16) and Discussion (Page 18).

Stronger evidence would be showing decoding of the compressed code during memory maintenance or recall, but this is not presented. On the contrary, during location recall (after the majority of memory maintenance is already over), colour decoding re-emerges, but in the un-compressed item-by-item code (Fig. 4B). The authors suggest that compression is consolidated at this point, but its utility at this late stage is not obvious.

Thank you for the important question we apologize for omitting previously - neural evidence for the compressive account.

The reason we did not perform neural decoding during maintenance is that previous EEG/MEG studies including our own failed to reveal robust and sustained time-resolved memory decoding during this period. This is posited to arise from “activity-silent” WM states, wherein memories are not necessarily retained in sustained firing but silently stored within connection weights of WM networks (Stokes, *Trends Cogn. Sci.,* 2015; Rose, *Curr Dir Psychol Sci*, 2020). Our previous work showed that by transiently perturbing the 'activity-silent' WM using a retrocue or neutral impulse, memories could be reactivated and robustly decoded from neural activities (Huang et al., eLife, 2021). However, due to the lack of transient events during retention in the current design, we do not expect robust decoding results during maintenance. As shown below (AB), this is indeed what we have observed, i.e., no robust neural decoding of trajectories during retention.

We further used alpha-band (8-11 Hz) neural activities, which have been shown to carry WM information (de Vries et al., Trends Cogn. Sci, 2020; Foster et al., Curr. Biol, 2016; Fukuda et al., J. Neurophysiol, 2016; Sutterer et al., PLOS Biol., 2019) to perform decoding analysis of compression trajectories during maintenance. As shown below, the alpha-band decoding results are indeed stronger than raw activities. Importantly, as shown below (CD), the aligned condition indeed showed significant and long-lasting decoding of compression trajectories (1st-2nd, 2nd-3rd) during retention, while the misaligned condition only showed decoding at the beginning (GH), which might be due to the non-specific offset response of the 3rd item. The results, although not as clear as those during encoding and recalling periods, support the reviewer’s hypothesis that the compressive strategy, if exploited, would be demonstrated during both encoding and maintenance periods. New results and related discussion have been added (Page 16, Supplementary Figure 4).

With regards to the observed item-by-item color replay during location recall, the reviewer was concerned that this was not consistent with the compressive account, given the lack of trajectory decoding.

First, item sequences stored in compressive formats need to be converted to sequences during serial recall. In other words, even though color and location sequences are retained in a compressive format (i.e., common 1st-2nd, 2nd-3rd trajectories) throughout the encoding and retention phases, they should be transferred to two sequences as outputs. This is exactly why we performed decoding analysis on individual color and location items rather than trajectories.

Second and most importantly, we observed serial replay of color sequences when recalling locations. In our view, these results constitute strong evidence for common structure, since the spontaneous color replay during location recall for aligned condition highlights the close bound between color and location sequences stored in WM. In fact, item-by-item serial replay has been well acknowledged as a critical neural index of cognitive maps, not only for spatial navigation but also for higher-order tasks (e.g., Liu et al., Cell, 2019; Liu et al., Science, 2021). Therefore, spontaneous color sequence replay during location sequence recall supports their shared underlying cognitive map.

Finally, spontaneous serial replay is also correlated with the reactivation of compressive trajectories during encoding (Supplementary Figure 3). This further indicates that serial replay during recalling is associated with memory reorganization formed during encoding.

Taken together, we posit that memories need to be converted to sequences as outputs, which leads to serial reactivations during recalling. Importantly, the observed spontaneous replay of color sequences for the aligned condition provides strong evidence supporting the associations between color and location sequences in WM.

We have now added relevant interpretations and discussions (Page 11&13).

**Reviewer #2 (Public Review):**
Summary:In this study, the authors wanted to test if using a shared relational structure by a sequence of colors in locations can be leveraged to reorganize and compress information.Strength:They applied machine learning to EEG data to decode the neural mechanism of reinstatement of visual stimuli at recall. They were able to show that when the location of colors is congruent with the semantically expected location (for example, green is closer to blue-green than purple) the related color information is reinstated at the probed location. This reinstatement was not present when the location and color were not semantically congruent (meaning that x displacement in color ring location did not displace colors in the color space to the same extent) and semantic knowledge of color relationship could not be used for reducing the working memory load or to benefit encoding and retrieval in short term memory.Weakness:The experiment and results did not address any reorganization of information or neural mechanism of working memory (that would be during the gap between encoding and retrieval).

We apologize for not presenting clear neural evidence for memory reorganization, particularly neural decoding during WM maintenance and retrieval, in the previous version. As below, we explain why the findings provide converging neural evidence for WM reorganization based on a shared cognitive map.

First, during the encoding phase when location and color sequences are serially presented, our results reveal reactivation of the 1st-2nd trajectories upon the onset of the 3rd item when location and color sequences are aligned with each other. The reactivation of 1st-2nd trajectory right after the emergence of 2nd-3rd trajectory for aligned but not for misaligned sequences strongly supports WM reorganization, since only stimulus sequences that could be compressed based on shared trajectories (aligned condition) show the co-occurrence of 1st-2nd and 2nd-3rd trajectories. Moreover, the relevance of 1st-2nd reactivation to behavioral measurements of color-location reorganization (i.e., behavioral trajectory correlation, Figure 5D) further indicates its link to WM reorganization.

Second, the reason we originally did not perform neural decoding during maintenance is that previous EEG/MEG studies including our own failed to reveal robust and sustained time-resolved memory decoding during this period. This is posited to arise from “activity-silent” WM states, wherein memories are not necessarily retained in sustained firing but silently stored within connection weights of WM networks (Stokes, *Trends Cogn. Sci.,* 2015; Wolff et al., Nat. Neurosci, 2017; Rose et al., *Curr Dir Psychol Sci*, 2020). Our previous work showed that by transiently perturbing the 'activity-silent' WM using a retrocue or neutral impulse, memories could be reactivated and robustly decoded from neural activities (Huang et al., eLife, 2021). However, due to the lack of transient events during retention in the current design, we do not expect robust decoding results during maintenance. As shown in Supplementary Figure 4(AB), this is indeed what we have observed, i.e., no robust neural decoding of trajectories during retention.

We then used alpha-band (8-11 Hz) neural activities, which have been found to carry WM information (de Vries et al., Trends Cogn. Sci, 2020; Foster et al., Curr. Biol, 2016; Fukuda et al., J. Neurophysiol, 2016; Sutterer et al., PLOS Biol., 2019) to perform decoding analysis of compression trajectories during maintenance. As shown below, the alpha-band decoding results are indeed stronger than raw activities. Importantly, as shown in Supplementary Figure 4(CD), the aligned condition indeed showed significant and long-lasting decoding of compression trajectories (1st-2nd, 2nd-3rd) during retention, while the misaligned condition only showed decoding at the beginning (GH), which might be due to the non-specific offset response of the 3rd item. The results, although not as clear as those during encoding and recalling periods, thus also support WM reorganization.

Finally, during the recalling period, we observed automatic serial replay of color sequences when recalling locations. In our view, these results constitute strong evidence for common structure, since the spontaneous color replay during location recall for aligned condition highlights the close bound between color and location sequences stored in WM. In fact, item-by-item serial replay has been well acknowledged as a critical neural index of cognitive maps, not only for spatial navigation but also for higher-order tasks (e.g., Liu et al., Cell, 2019; Liu et al., Science, 2021). Therefore, spontaneous replay of color sequence during location recall supports their shared underlying cognitive map. Moreover, the spontaneous serial replay is correlated with the reactivation of compressive trajectories during encoding (Supplementary Figure 3). This further indicates that serial replay during recalling is associated with memory reorganization formed during encoding.

Taken together, we have added updated results about the maintenance period (Page 16, Supplementary Figure 4) and included clarifications and interpretations about why the findings during the encoding and retrieval periods support the WM reorganization view (Page 15-16).

There was also a lack of evidence to rule out that the current observation can be addressed by schematic abstraction instead of the utilization of a cognitive map.The likely impact of the initial submission of the study would be in the utility of the methods that would be helpful for studying a sequence of stimuli at recall. The paper was discussed in a narrow and focused context, referring to limited studies on cognitive maps and replay. The bigger picture and long history of studying encoding and retrieval of schema-congruent and schema-incongruent events is not discussed.

We agree with the reviewer that cognitive map referred here could be understood as schematic abstraction. Cognitive map refers to the internal representation of spatial relations in a specific environment (Tolman 1948). Schematic abstraction denotes a more broad range of circumstances, whereby the gist or structure of multiple environments or episodes can be integrated (Bartlett, 1932; Farzanfar et al., *Nat. Rev. Neurosci*, 2023).

In other words, schema refers to highly abstract framework of prior knowledge that captures common patterns across related experiences, which does not necessarily occur in a spatial framework as cognitive maps do. Meanwhile, in the current design, we specifically manipulate the consistency of spatial trajectory distance between color and location sequences. Therefore, we would argue that cognitive map is a more conservative and appropriate term to frame our findings.

Relevant discussions have been added (Page 3&19).

We apologize for the lack of more generalized discussion and have added schema-related literatures. Thanks for the suggestion.

**Recommendations for the authors:**

**Reviewer #1 (Recommendations For The Authors):**
(1) Do time-frequency-domain data (e.g., alpha-band power) in the delay provide evidence for delay-period decoding of trajectory lengths? This might strengthen the case for compression.

Thanks for the suggestion. We now performed decoding analysis of the delay period based on alpha-band power. As shown in supplementary figure 4, both the 1st-2nd and 2nd-3rd trajectories could be decoded for the aligned condition.

Added in supplementary figure 4 and Page 16.

(2) Do participants erroneously apply the compression strategy in the misaligned condition? This would not show up in the trajectory error correlation analysis, but might be visible when examining correlations between raw trajectory lengths.

Thanks for raising this interesting suggestion. To test the hypothesis, we chose a typical misaligned condition where 1st-2nd trajectory distances are same between location and color sequences, while the 2nd-3rd trajectory distances are different between the two features.

In this case, participants might exploit the compression strategy for the first two items and erroneously apply the strategy to the 3rd item. If so, we would expect better memory performance for the first two items but worse memory for the 3rd item, compared to the rest of misaligned trials. As shown below, the 1st-2nd aligned trials showed marginally significant higher performance than misaligned trials for the first two items (t(32) = 1.907, p = 0.066, Cohen’s d = 0.332) . Unfortunately, we did not find significant worse performance for the 3rd item between the two conditions (t(32) = -0.4847, p = 0.631, Cohen’s d = -0.084). We observed significant interactions between the last two items and the alignment effect (t(32) = 2.082, p = 0.045, Cohen’s d = 0.362), indicating a trend of applying wrong compression strategy to the 3nd item.

**Author response image 1. sa2fig1:** 

(3a) Some more detail on some of the methods might help readers. For instance, did trajectories always move in a clockwise direction? Could the direction reverse on the third item? If not, did this induce a response bias? Could such a bias possibly account for the trajectory error correlations

Sorry for the unclear statement. For individual trial, both the color and location features of the three items are randomly selected from nine possible values without any constraint about the directions. That is to say, the trajectories can move in a clockwise or anticlockwise direction, and the direction can also reverse on the third item in some trials. Thus, we think the current design can actually help us to reduce the influence of response bias. Taking a step back, if trajectory error correlations are due to response bias, we should expect consistent significant correlation for all conditions, instead of only observing significant correlation for 1st-2nd and 2nd-3rd trajectories but not for 1st-3rd trajectory and only in aligned trajectory condition but not in misaligned condition. Therefore, we think the trajectory error correlations cannot be simply explained by response bias.

Details have been added (Page 23).

(3b) Is the colour wheel always oriented the same way for a participant? If so, given there are only nine colors, it seems possible that colors are mapped to locations and remembered in a location code instead. This does not seem to be a problem in principle for the behavioural findings, but might change the interpretation of what is being decoded from the EEG. If this is a possibility then this might be acknowledged.

The color wheel is always oriented the same way for each participant. We agree with the reviewer that it is possible that participants tend to map colors to locations and remembered in a location code. We don’t have sufficient evidence to rule out this possibility. One possible way could be running another experiment with varied color wheel during response period. Meanwhile, we would like to point out that the underlying logic of the current design is based on the facts that thinking spatially is intuitive and spatial metaphors like “location” and “distance” is commonly used to describe world, e.g., the well-known mental number line (Dehaene et al., *JEP: General,* 1993). Therefore, we expected participants to associate or integrate location and color maps based on trajectory distance.

The reviewer is correct that the color decoding would reflect spatial location rather than the genuine color feature. This is actually the point of the experimental design, whereby two irrelevant features could be possibly combined within a common cognitive map. Without the realignment of the two feature maps defined in space, subjects could not at all form the strategy to compress the two sequences. In other words, decoding of color sequences could be understood as neural representation of a series of corresponding locations along the ring that are independent of the physical locations of the items.

Interpretations and clarifications have been added (Page 23&26).

(4) Does the discretisation of the stimulus distribution (to only 9 possible locations) make the compression strategy easier to use? If the features had been continuously distributed across the location/colour circle, would participants still pick up on and use the shared trajectory structure?

Thanks for the question. Without further data, it’s hard to say whether the discretization of the stimulus distribution would make the compression strategy easier to use or not, compared to continuous distribution. Both outcomes seem possible. On the one hand, discrete stimulus distribution would result in discrete trajectory distribution, which helps participants to realize the common trajectory strategy. On the other hand, discrete stimulus distribution would result in category or label representation, which may weaken the effectiveness of structure compression strategy. We postulate that our findings could be generalized to continuous trajectories in a cognitive map within certain resolution.

(5a) Minor point: I disagree that avoiding the same points for location and colour for a given item allows them to be independently decoded. I would argue the contrary - this kind of constraint should create a small anti-correlation that in principle could lead to spurious decoding of one variable (although this seems unlikely here).

We appreciate the concern. As mentioned above, with discrete stimulus distribution (9 possible values for both color and location domains), it is quite possible that a fraction of trials would share same values in location and color. Therefore, the neural decoding for one domain might be confounded by another domain. To dissociate their neural representations, we imposed constraints that color and location could not occupy the same value for a given item.

We agree that this kind of constraint might create a small anti-correlation, even though it is not observed here. Future studies using continuous stimulus distribution would reduce the correlation or anti-correlation between stimuli.

(5b) Very minor point: 1,000 permutations for significance testing seems on the low side. Since some of the p-values are close to 0.05 it may be worth running more permutations.

Thanks for this suggestion. We got similar results using 1000 or 10000 permutations.

(6) Missing reference: H. H. Li et al., 2021 (line 213) seems not to be on the list of references.

Sorry for the mistake. Added.

**Reviewer #2 (Recommendations For The Authors):**
The study aimed to discuss the working memory mechanism, instead, it seems to be focused on the encoding and recall strategies after a short while, I recommend updating the manuscript to refer to the relevant cognitive mechanism.There was a strong voice on the effect of using the cognitive map in working memory, without any tests on if indeed a cognitive map was used (for example the novel link between stimuli and how a cognitive map can be used to infer shortcuts). Was the participant required to have any mental map beyond the schema of the shown color ring?In the current experiment, to discuss if the effect is driven by utilizing a cognitive map or schematic abstraction of color-relatedness, further analysis is required to possibly assess the effects of schema on neural activity and behavior. Namely,(1) Was there any reinstatement of schematically congruent (expected) colors that were probed by location 1, at locations 2 and 3 in the MAT condition?

Thanks for pointing out this possibility. However, we don’t think there will be stable color expectations given location information under the MAT condition. First, as the trajectory distance varied on a trial-by-trial basis, no prior common trajectory knowledge could be used to make inference about the current stimuli in individual trial. Second, the starting points for color and location (1st item) were randomly and independently selected, such that color sequence could not be predicted based on the location sequence for both aligned and misaligned conditions.

(2) Given that response time can be a behavioral marker of schematic conflict, was the response time faster for congruent than incongruent conditions?

Thanks for this question. Unfortunately, due to the experimental design, the response time could not be used as a behavioral marker to infer mental conflicts, since participants were not required to respond as fast as possible. Instead, they took their own pace to reproduce sequences without time limit. They could even take a short break before submitting their response to initiate the next trial.

(3) In case you cannot rule out that utilizing schema is the cognitive mechanism that supports working memory performance (the behavior), please add the classical literature (on the memory of schematically congruent and incongruent events) to the discussion.

Thanks for this suggestion and we have added relevant literatures now (Page 3&19).

(4) On page 6, 'common structure in the cognitive map' is the schema, isn't it?

Correct. Based on our understanding, ‘common structure in the cognitive map’ is a spatial schema.

(5) In Figure 2 EFG, would you please use a mixed effect model or show evidence that all participants demonstrated a correlation between the location trajectory error and color trajectory error?

Thanks for the suggestion. We have added the mixed effect model results, which are consistent with Figure 2EFG (AT: 1st-2nd trajectory, β = 0.071, t = 4.215, p < 0.001; 2nd-3rd trajectory, β = 0.077, t = 3.570, p < 0.001; 1st-3rd trajectory, β = 0.019, t = 1.118, p = 0.264; MAT: 1st-2nd trajectory, β = 0.031, t = 1.572, p = 0.116; 2nd-3rd trajectory, β = 0.002, t = 0.128 , p = 0.898; 1st-3rd trajectory, β = -0.017, t = -1.024, p = 0.306).

In general, doesn't such correlation just show that good participants/trials were good (some did well in the study and some did poorly throughout?)

We don’t think the trajectory error correlation results just reveal that some participants did well and some participants did poorly. If that is the case, we shouldn’t observe significant correlation in Figure 2D, where we first run correlation for each participant and then test correlation significance at group level. Indeed, trajectory error correlation between color and location domains characterizes the consistent changes between the two domains.

It is worth to note that the correlation was estimated with signed trajectory errors in color and location domains, which meant that we indeed cared about whether the errors in the two domains were consistently varied in the same direction, i.e., whether longer trajectory memory compared to the actual trajectory in location domain would predict longer trajectory memory in color domain.

Moreover, as shown in Figure 2EFG, by dividing trials into 4 bins according to the location trajectory error for each participant and pooling the data across participants, we observed 4 clusters along x-axis (location trajectory error). This suggests that participants’ memory performance is rather consistent instead of being extremely good or bad. Besides, if trajectory error correlation is due to different overall memory performance between participants, we should observe significant trajectory error correlations both in AT and MAT conditions, instead of only under AT condition and for 1st-2nd and 2nd-3rd trajectories but not for 1st-3rd trajectory.

In Figure 2 G, is the marginal error just too big to be sensitive? I am not sure what we are learning here, please clarify.

Sorry for the confusion. To examine this possibility, we excluded errors which are beyond 2.5 * σ, and still observed non-significant 1st-3rd trajectory error correlation between color and location domains (r = 0.119, p = 0.167).

The 1st-3rd trajectory showed nonsignificant behavioral correlation and neural representation, which suggests that the current sequential memory task would encourage participants to organize all information by relying more on the adjacent items and their distance. Thus, we think the 1st-3rd trajectory would serve as a control trajectory, which helps us not only exclude other possible explanation (e.g., systematic response bias), but also validate current findings both in behavioral and neural level.

Results and statements (Page 10-11) added now.

(6) Regarding the first lines on page 11, did you do qualitative research to know if less information was encoded in congruent conditions?

The current experimental design is inspired by the mental compression of spatial sequence studies from Dehaene’s lab (Amalric er al., 2017; Roumi et al., 2021), in which they propose that human brain compresses spatial sequence using an abstract language and formalize minimal description length of a sequence as the “language-of-thought complexity.” Based on this evidence, we think less information is required to describe congruent condition compared to incongruent condition. This idea is supported by better memory performance for congruent condition. Unfortunately, we couldn’t manage to quantify how less information was encoded in congruent condition.